# Properties and Microstructures of Crushed Rock Based-Alkaline Activated Material for Roadway Applications

**DOI:** 10.3390/ma15093181

**Published:** 2022-04-28

**Authors:** Teewara Suwan, Peerapong Jitsangiam, Hemwadee Thongchua, Ubolluk Rattanasak, Thanon Bualuang, Phattharachai Maichin

**Affiliations:** 1Center of Excellence in Natural Disaster Management, Department of Civil Engineering, Faculty of Engineering, Chiang Mai University, Huay Kaew Road, Mueang, Chiang Mai 50200, Thailand; teewara.s@cmu.ac.th (T.S.); thanon_bualuang@cmu.ac.th (T.B.); phattharachai_mai@cmu.ac.th (P.M.); 2Graduate Program in Civil Engineering, Department of Civil Engineering, Faculty of Engineering, Chiang Mai University, Huay Kaew Road, Muang, Chiang Mai 50200, Thailand; hemwadee_th@cmu.ac.th; 3Department of Chemistry, Faculty of Science, Burapha University, Muang, Chonburi 20131, Thailand; ubolluk@go.buu.ac.th

**Keywords:** alkaline activated material, cement-stabilized pavement, crushed rock, low strength construction material, microstructure

## Abstract

The worldwide demand for roads to serve global economic growth has led to the increasing popularity of road improvement using cement. This, in turn, has led to increased demand for cement and the associated problem of CO_2_ emissions. Alkaline-activated materials (AAMs) could be an alternative binder for relatively low strength construction and rehabilitation as a cement replacement material. Compared to other applications, the lower strength requirements of road construction materials could ease any difficulties with AAM production. In this study, crushed rock (CR) was used as a prime raw material. The mechanisms and microstructures of the hardened AAM were investigated along with its mechanical properties. The results showed that CR-based AAM with an optimum mixture of 5 M of NaOH concentration, an SS/SH ratio of 1.00, and a liquid alkaline-to-binder (L/B) ratio of 0.5 could be used for roadway applications. At this ratio, the paste samples cured at room temperature (26 ± 3 °C) had an early compressive strength (3 days-age) of 3.82 MPa, while the paste samples cured at 60 °C had an early compressive strength of 6.45 MPa. The targeted strength results were able to be applied to a cement-treated base (CTB) for pavement and roadway applications (2.1 to 5.5 MPa).

## 1. Introduction

Recently, with increased attention paid to global warming, the roadway construction sector has been pressured to be more sustainable, i.e., to develop clean and green technologies. Innovative and sustainable materials based on the concept of alkaline-activated materials (AAM) have recently been introduced to the sector. They are in the early stages of investigation [1,2,3]. AAM is the broadest classification of binder systems derived by a chemical reaction of alkaline sources (in solid or dissolved form) with any prime materials [4,5]. This prime material can be a calcium silicate compound in ordinary Portland cement or a more aluminosilicate-rich precursor (e.g., fly ash, bottom ash, metallurgical slag, natural pozzolan). AAM is believed to be a green construction material for the future because it produces significantly less CO_2_ than ordinary Portland cement (OPC). OPC is the most frequently used construction material, and it is the most popular stabilizing agent [6]. AAM could yield sufficient strength and other satisfying parameters, resulting from the chemical reaction between an alkaline solution and a solid aluminosilicate precursor [7,8,9,10,11,12,13]. However, alkaline-activated cement technology has apparent limitations. Heat curing is a primary factor of the complete chemical reaction processes, and has proven to be a significant problem. The difficulty of heat curing hinders its acceptance as an OPC replacement, among other applications. Adopting an AAM with low strength for specific purposes, e.g., unfired bricks or roadway applications, has been a challenge in developing new green construction materials. This study gave a preliminary introduction to relatively low strength AAM as an environmentally friendly road construction material to replace OPC. This could help to minimize global CO_2_ emissions by the cement production industry.

In the manufacturing industry, cement production emits approximately 8% of global CO_2_ emissions [14]. Every ton of cement production emits approximately one ton of CO_2_ into the atmosphere [15]. OPC is the most widely used material for many kinds of structures. In order to mitigate the use of OPC, cement substitute materials, such as AAMs, have been introduced [16,17,18]. AAMs have the potential to replace OPC and reduce CO_2_ emissions. It has been claimed that AAMs emit 80% less CO_2_ compared to OPC [19]. The AAM can be synthesized by mixing byproduct materials (e.g., fly ash, bottom ash, or furnace slag) with alkaline solutions (e.g., sodium hydroxide and sodium silicate) [20,21,22,23,24]. Figure 1 shows the synthesis process of AAM based on the view of this study. The main composition of AAM, based on calcium content, was calcium aluminosilicate hydrate (C-A-S-H). It had a layer structure similar to calcium silicate hydrate (C-S-H) and sodium aluminosilicate hydrate (N-A-S-H), a two-to-three-dimensional structure of Si-O-Al in the matrices [20,25]. Alkaline activation has two phases (high-calcium alkaline activated and low-calcium alkaline activated), otherwise known as geopolymers [26].

The targeted applications of this relatively low-strength AAM may be applied to the roadway with crushed rock as a precursor. Road pavement has an average design life of about 15–30 years [27,28]. Hence, the number of roads that require rehabilitation and reconstruction has grown to meet the high demand for transporting goods [29] and people under the current economic circumstances. Therefore, reconstructing and rehabilitating road pavements in an environmentally sustainable and cost-effective manner is an engineering challenge [30]. Typically, road pavement requires a relatively low-strength material compared to other civil engineering structures. To improve the mechanical properties of road base materials, cement stabilization through in situ pavement recycling has been the most popular technique. In this approach, a small amount of cement is added to existing pavement materials. There is no transportation of new road pavement construction materials required [31,32]. In situ cementitious rehabilitation (i.e., pavement recycling with cement stabilization for road pavements, as shown in Figure 2) is considered a cheaper (35–50% of the cost of reconstruction) and potentially environmentally sustainable solution because it recycles existing pavement materials [33]. It was found that the compressive strength of cement-stabilized base materials increased with higher cement content [34]. Currently, OPC is used for more than 80% of road rehabilitation activities worldwide [35]. Although pavement rehabilitation uses only about 2–8% (by weight) of OPC compared to 10–16% for concrete, the amount of concrete utilized is relatively small compared to the pavements in need of rehabilitation. When in situ pavement rehabilitation is more widely used, as predicted by the recent growth of road networks worldwide, OPC use for this purpose could vastly increase (estimated to be about 150 tons/km) [36]. This higher amount of cement use would cause relatively sharp increases in CO_2_ emissions owing to cement production. Therefore, alkaline-activated stabilizers (binders) for in situ pavement rehabilitation should be cleverly engineered as a potential environmentally and structurally sustainable solution. 

The primary raw material used in this study was crushed rock (CR), a calcium-rich material. The presence of calcium in the AAM system can provide the basis for gaining strength at room temperature [37,38,39,40]. This study ultimately aimed to explore the possibility of AAM as a cementitious material using CR as source material with no other active precursors (e.g., fly ash). The influential factors of AAM for road stabilization were investigated through compressive strength. Thus, the results introduced the possible use of a crushed rock-based AAM with relatively low strength and a more environmentally friendly construction material.

In this study, CR, which is generally used as a road base material, was used as the source material of AAM production. The physical properties of AAM were studied, including setting times, flowability, and compressive strength. Results were compared with the requirements for roadway materials. Additionally, the chemical compositions, morphology, and microstructure of AAM were analyzed using X-ray fluorescence (XRF), X-ray diffraction (XRD), and a scanning electron microscope (SEM), respectively.

## 2. Materials

### 2.1. Raw Materials (Crushed Rock, CR)

The CR used as a raw material for AAM production was obtained from a Chiang Mai local quarry of Sila Mae Tha Co., Ltd., Chiang Mai, Thailand. It was crushed into smaller sizes and sieved with mesh sieve no. 100, obtaining a fine material. The chemical compositions of the CR, determined by energy-dispersive X-ray fluorescence (EDXRF) analysis, are shown in Table 1. The average particle size of the CR, determined by dynamic light-scattering particle size analyzer (Nanosizer-DLS), was 50.2 μm. A total of 80% of CR particles were smaller than 90.7 μm, while their specific surface area was 0.2573 m^2^/g (Figure 3).

### 2.2. Alkaline Activators

The alkaline materials used in this study were sodium hydroxide (NaOH) and sodium silicate (Na_2_SiO_3_). NaOH solution was prepared by dissolving sodium hydroxide pellets in purified water, obtaining concentrations of 3, 5, and 8 molars (M). The solution was allowed to cool down to room temperature. Sodium silicate solution was acquired from Union Science Co., Ltd., Muang, Chiang Mai, Thailand, with sodium oxide (Na_2_O) and silicon oxide (SiO_2_) contents of 16.3% and 34.2% by weight, respectively.

## 3. Experimental Programs and Analytical Methods

### 3.1. Experimental Programs and Sample Preparations

The mixture designations of CR-AAM are presented in Table 2. Figure 4 shows the framework and methodology of this study. First, fine-grained CR was analyzed for its chemical composition and morphology using XRF and XRD analysis. The main parameters for the study were the NaOH concentration, a mass ratio of sodium silicate/sodium hydroxide (SS/SH), and a mass ratio of liquid alkaline/binder (L/B). Two curing conditions were studied: room temperature (RT) curing and oven curing. CR and the specific amount of prepared alkaline solutions were initially mixed for two minutes. After 30 s, time was paused to remove some adhered paste from the middle of the mixing pot, then mixing was restarted for another two minutes. After thorough mixing, the freshly blended CR-based AAM paste was neatly cast into the prepared molds and wrapped with a plastic sheet to prevent moisture loss. For oven curing, the samples were placed in the oven at 60 °C for 24 h. Then, the samples were removed from the oven and allowed to cool down at room temperature (26 ± 3 °C). After 24 h of casting, all samples were demolded and wrapped with a plastic sheet and kept at room temperature until reaching the testing age. It is noted that the preparation methods and experimental factors were carried out following the previous research study and literature [34]. 

### 3.2. Analytical Methods

Setting times for the fresh CR-based AAM paste were determined using a Vicat apparatus with ASTM C191-a18 [41], while flow tests were carried out in accordance with ASTM C230/C230M-14 [42]. The compressive strength tests were performed on a 250 kN Control universal testing machine (UTM) by using 40 mm × 40 mm × 160 mm prism specimens with British standard EN 196-1 [43]. The samples were tested at 3 days and 28 days. The chemical composition of CR was determined by using XRF energy-dispersive X-ray fluorescence (EDXRF), Epsilon X-ray fluorescence (XRF) spectrometer, with PANalytical’s EDXRF platform system, Malvern Panalytical Company. A 4-circle kappa goniometer X-ray diffraction (XRD) machine with a microfocus sealed tube (Mo) and direct photon counting detector (HyPixBantam) was used to analyze the structural formation of the resulting products, along with the Cambridge Structure Database (CSD). Microstructures of the hardened samples were analyzed by SEM JSM-IT300 scanning electron microscope (SEM), JEOL JSM-5910LV, with 30 kV vacuum mode on the lanthanum hexaboride cathodes and a minimum resolution of 2.0 nm.

## 4. Results and Discussion

### 4.1. XRD Analysis

XRD analysis was carried out to investigate the structural formations and chemical compounds of both original crushed rock material and AAMs. For this study, 5 M NaOH with an L/B ratio of 0.50 and SS/SH ratio of 1.00 were set. The XRD patterns of the original crushed rock revealed a structure formation of quartz (Q), calcite (C), and portlandite (P) in the forms of SiO_2_, CaCO_3,_ and Ca(OH)_2_, respectively. Aluminum oxide (Al_2_O_3_) and silicon dioxide (SiO_2_) were also found in different forms (* -Na, Al, Ca, Si compound). Hematite (H) at the small peak around 31.1 two-theta degrees referred to iron compounds in the raw material (Fe_2_O_3_). It was noted that a sharp peak of calcite located at 29.4 two-theta degrees referred to the rampant amount of calcium content in the crushed rock material; they were transformed into calcium silicate hydrate (C-S-H) after polymerization/hydration processes. Goethite (G), a form of soil or other low-temperature sediments, was observed as FeO(OH) or FeAl_2_O_4_ at the same peak of hematite around 31.1 two-theta degrees. Additionally, Portlandite (P), or Ca(OH)_2_, was found in both curing conditions due to the reaction of calcium compounds and hydroxide ions (OH^−^) in the systems. The quartz (Q) and aluminum compound in the original crushed rock, at 26.6 and 45.7 two-theta degrees, could be transformed to Kyanite (K), an aluminosilicate compound (Al_2_O_5_Si) in the structure. The graphs mainly show the sharp peaks, which indicate the formation of crystallinity. However, small humps of those alkaline-activated mixtures were also observed between 28 and 34 two-theta degrees, indicating the partial formation of amorphous phases of the alkaline-activated structure (Figure 5).

Comparative XRD patterns of AAMs with both curing conditions were also carried out at 28 days, as shown in Figure 6 and Figure 7. There were no significant differences in XRD results between curing at room temperature and 60 °C; the patterns were similar. Nevertheless, higher strengths were achieved using oven curing as compared to room temperature curing. Oven curing provided more polymerization, with a more robust polymeric chain and dense matrices.

Generally, a higher concentration of NaOH could provide a higher dissolution rate for alkaline activated pastes to re-polymerize their structure [25]. Greater intensity of kyanite (K) at 26.7 two-theta degrees was observed, with NaOH concentrations of 8 M, 5 M, and 3 M. Similar characteristics were observed at the peak around 29.5 two-theta degrees of calcium silicate hydrate (C-S-H, C’) as mixtures with higher NaOH concentrations gained more intensity from XRD analysis. In some cases, the crystalline form of calcium aluminosilicate hydrated (C-A-S-H) was also detected from CaO, Al_2_O_3_, and SiO_2_ in the parent raw material. A high curing temperature (i.e., 60 °C) could increase the formation rates of both C-S-H and C-A-S-H, resulting in the early development of compressive strength [44,45,46,47]. The chemical formula, from the XRD database, of C-S-H (Tobermorite) and C-A-S-H were Ca_5_Si_6_O_16_(OH)_2_·4H_2_O and CaAl_2_Si_2_O_8_·4(H_2_O), respectively.

The peak area of the XRD pattern was also calculated; a two-theta degree in the range of 25 to 35 was selected, as reaction products mainly occurred in this range. Results of the peak area are shown in Table 3. As shown, the peak area increased with high NaOH concentration; it increased by 10–20% of the peak area, compared to 3 M NaOH. Oven curing resulted in a higher peak area than RT curing, as heat activated more product formation. The results for peak area corresponded to compressive strength.

### 4.2. Setting Times and Flowability

Generally, in the construction field, a concrete mixture should be operated for a reasonable time, i.e., neither a fast nor a slow set. This property allows the concrete mixture to have a time gap for transportation, pouring, and compacting. It should be deformed within a specific period to accommodate working schedules [48]. Setting time is the first step to verify the setting behavior of the fresh paste, which commonly depends on chemical reactions in the raw materials [49,50]. The mixtures of the CR-AAM paste were prepared with a constant liquid alkaline/binder (L/B) ratio of 0.55 for the setting time and flow test. SS/SH ratios were used at 0.67, 1.00, and 1.50, while the respective NaOH concentrations were 3, 5, and 8 M. Results were shown in Figure 8 and Figure 9. The initial setting time of the paste complied with the standards of OPC, i.e., not less than 45 min for 3 M and 5 M NaOH usage. A rapid setting was observed with 8 M NaOH usage since the high concentration reacted quickly with the calcium compound in the system. The trend of setting times showed that an increase in the SS/SH ratio from 0.67 to 1.50 delayed the formation of the pastes due to the appearance of many silicon ions from the sodium silicate [51]. It has been reported that a higher SS/SH ratio could delay the initial setting time and affect the strength development after the hardening process [34,52,53].

The use of higher NaOH concentration led to more leaching of silica and alumina from the CR. It could react rapidly with the calcium compound in the CR, providing calcium hydroxide (C-H), calcium silicate hydrate (C-S-H), and calcium aluminosilicate hydrated (C-A-S-H) for fast setting behavior [7].

Flowability is the ability of freshly mixed cement or concrete to move or operate in practical works. In concrete operation, the flow value is known as workability. In general, high water content provides excellent workability, which is preferred for onsite applications. However, too much water decreases the compressive strength and causes more air voids in the hardened cement structure [54,55]. Consequently, a flow test was carried out to determine the appropriate mixtures and constituents for crushed rock-based alkaline-activated pastes. The mixes for the flow test were the same as those used for investigating setting times.

The results were in line with previous research studies [56,57]. The flow values decreased when the concentration of NaOH increased from 3 to 8 M. This was directly related to the overall water content in the mixtures, as higher NaOH concentrations commonly provide less flowability. Additionally, C-H, C-S-H, and C-A-S-H were formed in the mixture. In contrast, the sodium silicate solution’s high viscosity was also caused by coagulated mixtures, leading to low flow values. Therefore, the higher SS/SH ratio of 1.50 resulted in lower flow values than those resulting from ratios of 1.00 and 0.67. According to the previous study [34], the optimal SS/SH ratio of 1.00 was a suitable value to achieve both mechanical strength and workability. This ratio was also set for curing the AAM at RT and 60, respectively. The summary results of the setting times and flowability of CR-AAM are presented in Figure 8 and Figure 9.

### 4.3. Compressive Strength and Microstructure Analysis

#### 4.3.1. Effects of NaOH Concentrations and Curing Conditions

The effects of NaOH concentrations on the strength of AAM were analyzed. An L/B ratio of 0.50 and an SS/SH ratio of 1.00 were set for curing of the AAM at RT and 60 °C. The results showed that the strength of all mixtures increased with longer curing times, as shown in Figure 10. There were approximately 40% and 20% increments, respectively, for 5 M and 8 M NaOH, for both curing conditions. The chemical reactions for high calcium, alkaline-activated materials occurred very quickly [52,53]. Strength varied in the range of 3.82 to 5.4 MPa for RT curing and 6.45 to 9.40 MPa for oven curing. Therefore, when considering compressive strength itself, the CR-AAM of this study, prepared with 5 M and 8 M NaOH, could be used in relatively low-strength applications, e.g., as road construction material. It was noted that there were 54 testing pieces for each RT curing and oven curing, with less than 10% of standard deviation.

SEM images in Figure 11 and Figure 12 show the microstructures of CR-AAM in both RT and oven curing conditions prepared with an SS/SH ratio of 1.00 and L/B ratio of 0.50. It was observed that a structural formation could be rapidly achieved at a very early age owing to the alkaline reaction of high calcium content in CR. Fast-setting behavior can be found in other raw material precursors with high calcium content e.g., clay, high calcium fly ash, or OPC-incorporated binder. In fact, calcium aluminosilicate hydrate (C-A-S-H) and sodium aluminosilicate hydrate (N-A-S-H) were reported as the main formations [20,25,26]. The fast setting could be one of the advantages of using this AAM in some applications, e.g., as a repairing material, a rapid demolding material, or as a precast component, among many others. Moreover, a drying and curing process at mild-to-medium temperatures (26 °C to 60 °C) could significantly improve the polymerization of AAM [20,37,58]. Microstructure patterns of both curing conditions were slightly different. The firm and dense microstructures were observed in oven-cured AAM, implying more thorough polymerization in the heat curing environment.

From the SEM results, it was summarized that an appropriate range of NaOH concentration for CR-AAM was 5 M to 8 M, because the required strength was achieved at both curing conditions. However, using a 5 M NaOH in the mixture yielded cost savings and greater workability when fresh paste for low-strength applications was produced.

#### 4.3.2. Effects of Sodium Silicate/Sodium Hydroxide Ratios and Curing Conditions

The effects of SS/SH ratios of 0.67, 1.00, and 1.50 on compressive strength and microstructures of CR-AAM were studied. Additionally, 5 M NaOH and L/B ratio of 0.50 were set for this study. Results are shown in Figure 13. A similar trend was observed compared with the effects of NaOH concentrations (Figure 10). The strength of all the mixtures was developed by the curing time, and high strength achievement occurred with oven curing at 60 °C.

With a constant 5 M NaOH, a higher SS/SH ratio provided an additional source of silica (Si) in addition to controlling the binding activity of the cementitious systems. The compressive strength of the AAMs cured at room temperature for 28 days increased from 4.79 MPa to 5.18 MPa when the SS/SH ratio increased from 0.67 to 1.00, whereas the compressive strength of AAMs cured at 60 °C for 28 days rose from 5.55 MPa to 9.40 MPa. However, the compressive strength fell—under both curing conditions—when the SS/SH ratio was 1.50 (Figure 13). A higher ratio of SS/SH commonly gains an inert characteristic of the solution. A viscous sodium silicate could retard the chemical reaction after mixing because of insufficient water in the system. In addition, the formation of silica gel and the existence of excess sodium silicate in this system could lower strength. It was noted that there were 54 testing pieces for each RT curing and oven curing, with less than 10% of standard deviation.

Figure 14 shows SEM images of the AAMs cured at room temperature with different SS/SH ratios. Although there were no significant differences in visual screening observations, at SS/SH ratios of 1.00 and 1.50, the microstructures of the mixtures seemed to be more compact. They had denser matrices than when the SS/SH ratio was 0.67. Thus, it was concluded that the optimal SS/SH ratio for AAM was 1.00. Similar outcomes were also reported by Jitsangiam (2019), who found that the optimal SS/SH ratio of 1.00 not only achieved maximum compressive strength but also provided appropriate practical characteristics in both setting behavior and flowability [34]. At this mixture proportion, the cement could achieve maximum compressive strength (at 5.18 MPa and 9.40 MPa for room temperature and oven curing, respectively, at 28 days) and also have sufficient flowability for practical work applications. Additionally, the material was obtained at a lower cost due to the reduced use of sodium silicate solution.

#### 4.3.3. Effects of Liquid Alkaline/Binder Ratios and Curing Regimes

The liquid alkaline/binder (L/B) ratio is an essential factor for AAM synthesis as it indicates the amount of overall water in the system. The L/B ratio is somewhat similar to the water/cement (w/c) ratio in Portland cement production. However, the amount of water in liquid alkaline (L) must be recalculated from the concentrated solution before comparing it to water (w) in any cementitious mixture. A water-to-solid (w/s) ratio, in this study, referred to the total mass of water in the sodium hydroxide solution and the sodium silicate solution, compared to the mass of all solids from crushed rock, micro-pearl sodium hydroxide, and sodium silicate solid. The w/s ratios of all mixtures are presented in Table 4.

The water/solid (w/s) ratios are plotted against liquid alkaline/binder (L/B) ratios in Figure 15. Properties of the final hardened products, as influenced by the amount of water, could be observed: (i) the w/s ratios decreased when the NaOH concentration increased, (ii) the w/s ratios decreased when the SS/SH ratio increased, and (iii) the w/s ratios increased when the L/B ratio increased.

The effects of L/B ratios of 0.45, 0.50, and 0.55 on the mechanical properties of CR-AAM material were studied. A NaOH concentration of 5 M and a SS/SH ratio of 1.00 were set for curing mixtures at room temperature and 60 °C in the oven. It should be noted that the compressive strengths of all the mixtures were also developed through the passing of time, as shown in Figure 16. It was noted that there were 54 testing pieces for each RT curing and oven curing, with less than 10% of standard deviation.

The highest compressive strengths were achieved for room temperature curing with an L/B ratio of 0.45, when aged for three days (4.48 MPa) and 28 days (6.88 MPa). The compressive strength levels were very close to each other when the L/B ratios were 0.50 and 0.55; they were lower than when the L/B ratio was 0.45. Additionally, the trend of compressive strength for mixtures cured at 60 °C was significantly different. The mixture obtained the highest compressive strength with an L/B ratio of 0.50, followed by 0.45 and then 0.55.

From these results, it was concluded that the maximum strength was obtained for curing at room temperature when the L/B ratio was 0.45. On the other hand, the greatest compressive strength for oven curing was achieved when the L/B ratio was 0.50. In this case, the additional water content was able to compensate for some evaporation from the heat curing process. Therefore, maximum compressive strength was achieved after curing for a more extended period. However, water in the AAM material primarily controlled the alkaline activator’s concentration. In different cases, low alkaline solution concentration with a suitable high L/B ratio could achieve superior properties due to felicitous conditions for the hardened products.

With an optimal NaOH concentration of 5 M and an SS/SH ratio of 1.00, the water content (w/s ratio) of CR-AAM material should be between 0.26 and 0.28 (equivalent to an L/B ratio between 0.45 and 0.50) to achieve both setting behavior and workability, as seen in Section 4.2. It was emphasized that the relatively low strength of CR-based AAM was obtained mainly due to the use of the original CR itself. Less reactive CR (in comparison to OPC, fly ash, or any calcined materials) could suffer from less complete crystallinity, as sharp peaks were observed in XRD patterns [58,59,60,61]. Very few broad humps, which indicate amorphous phases, were found, due to less SiO_2_ and Al_2_O_3_ in the CR precursor [62,63]. However, the achieved compressive strengths surpassed the target compressive strength of the cement treat base for pavement or unfired brick, between 2.1 MPa and 7.0 MPa. 

## 5. Concluding Remarks

This preliminary study on the properties and microstructures of CR-AAM was carried out to explore opportunities to replace OPC and develop an alternative, environmentally friendly construction material. The conclusions were:The setting time of crushed rock-based alkaline-activated paste depended mainly on the concentration of NaOH. Rapid setting occurred at a concentration of 8 M, as the high concentration reacted quickly with calcium in the system. Flow values decreased when the concentration of NaOH increased from 3 M to 8 M, as less water was present. The same results occurred when a higher SS/SH ratio was applied.The compressive strength of crushed rock-based alkaline-activated paste was developed with the passing of time. Moreover, it was clear that heat curing (oven curing at 60 °C) led to higher compressive strengths than curing at room (ambient) temperature. The relatively low compressive strength of CR-based AAM was primarily due to the less reactive CR, in comparison to OPC, fly ash, or any calcined materials. However, the room temperature curing process could lead to the low-strength alkaline-activated cement required for low-strength applications, e.g., roadway, pavement, or construction blocks.NaOH concentrations between 5 M and 8 M gave the materials a satisfying strength. However, a 5 M concentration was preferable for setting time, flowability, and costs. An SS/SH ratio of 1.00 provided the highest compressive strength and flowability. For room curing, an L/B ratio of 0.45 yielded the best maximum strength value for practical work applications. The best L/B ratio for oven curing was 0.50, to compensate for evaporation during the heat curing process.

Overall, this preliminary investigation of crushed rock-based AAM indicated that the best mixture had a NaOH concentration of 5 M, an SS/SH ratio of 1.00, and an L/B ratio of 0.50. With these parameters, paste cured at room temperature yielded an early compressive strength of 3.82 MPa, and paste cured at 60 °C yielded an early compressive strength of 6.45 MPa. These results surpassed the target compressive strength of cement treat base (CTB) for pavement or unfired brick, which is between 2.1 MPa and 7.0 MPa.

## Figures and Tables

**Figure 1 materials-15-03181-f001:**
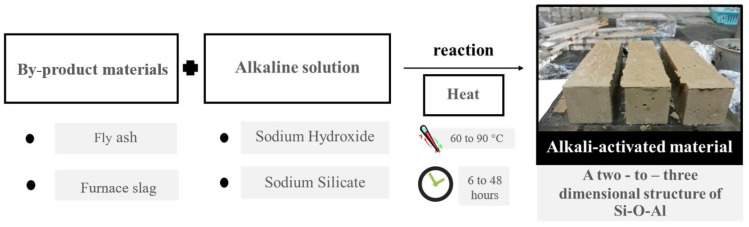
A typical process used to synthesize alkaline-activated materials.

**Figure 2 materials-15-03181-f002:**
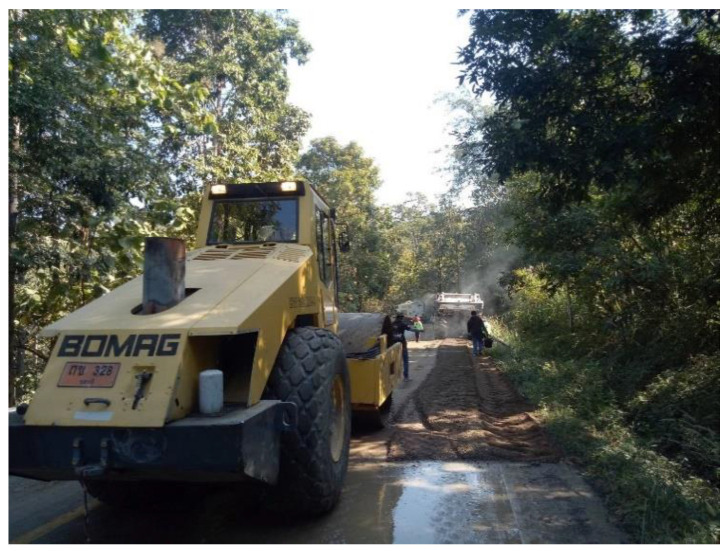
Possible application using AAM for in situ pavement recycling, Mae Hong Son province, Northern Thailand (author).

**Figure 3 materials-15-03181-f003:**
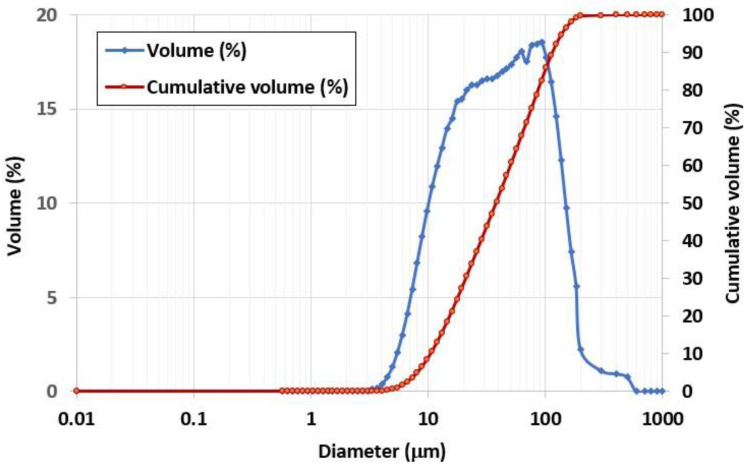
Particle size distribution of CR particles.

**Figure 4 materials-15-03181-f004:**
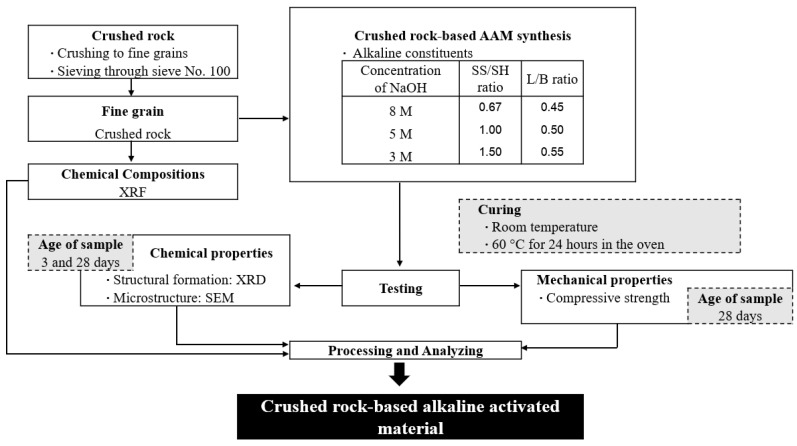
Experimental programs.

**Figure 5 materials-15-03181-f005:**
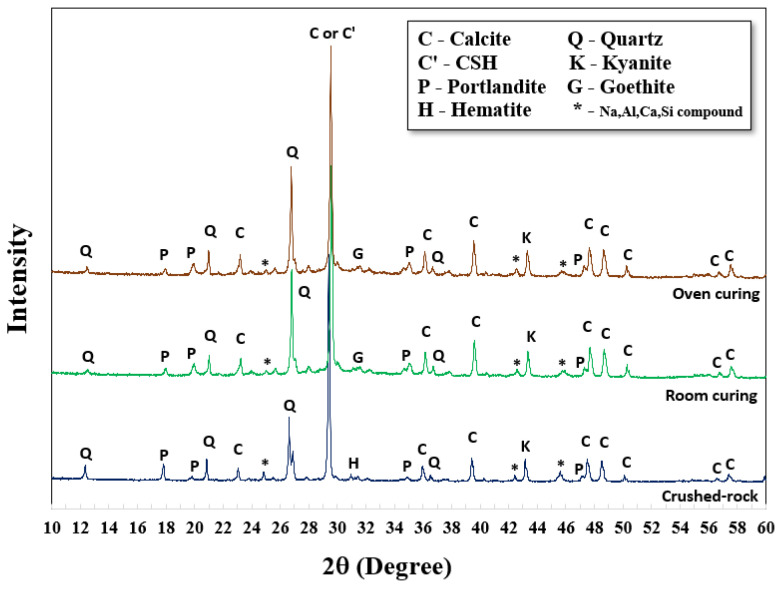
XRD patterns of original CR and AAM prepared with 5 M NaOH at 28 days.

**Figure 6 materials-15-03181-f006:**
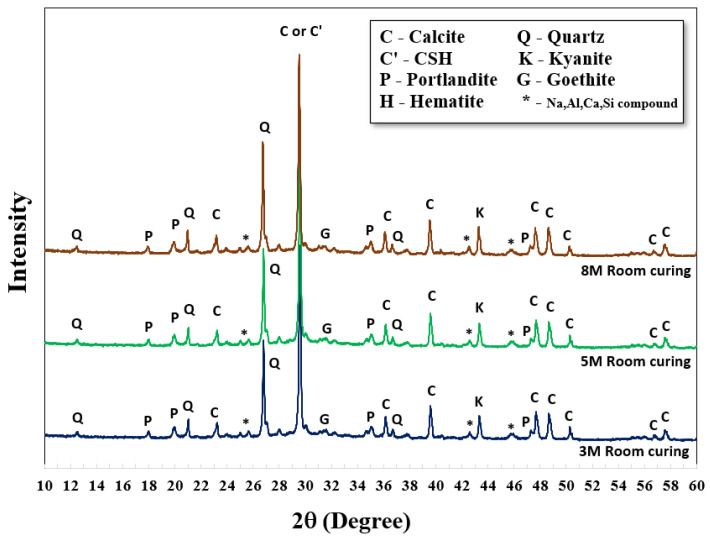
XRD patterns of AAM cured at room temperature.

**Figure 7 materials-15-03181-f007:**
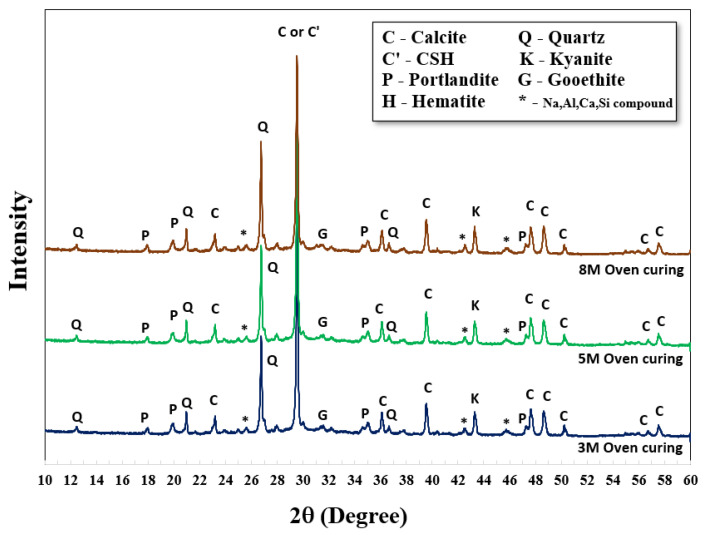
XRD patterns of AAM cured in the oven at 60 °C for 24 h.

**Figure 8 materials-15-03181-f008:**
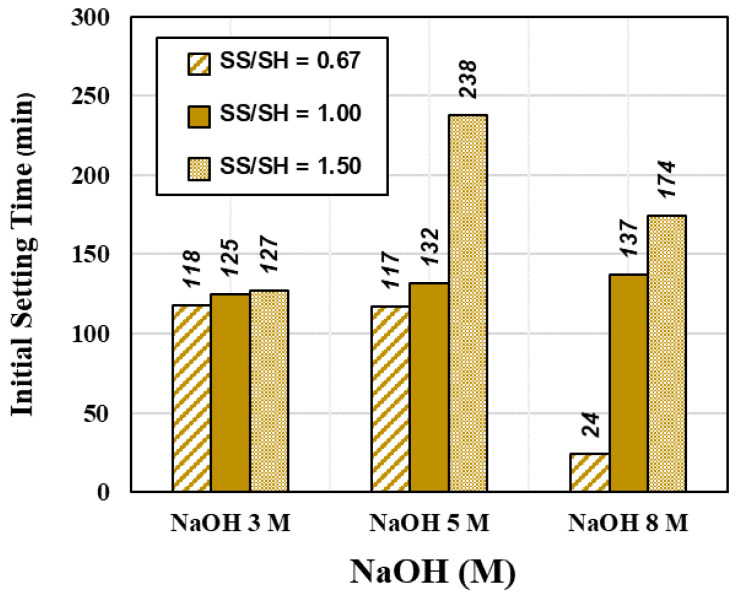
Setting time of various NaOH concentrations and SS/SH ratios.

**Figure 9 materials-15-03181-f009:**
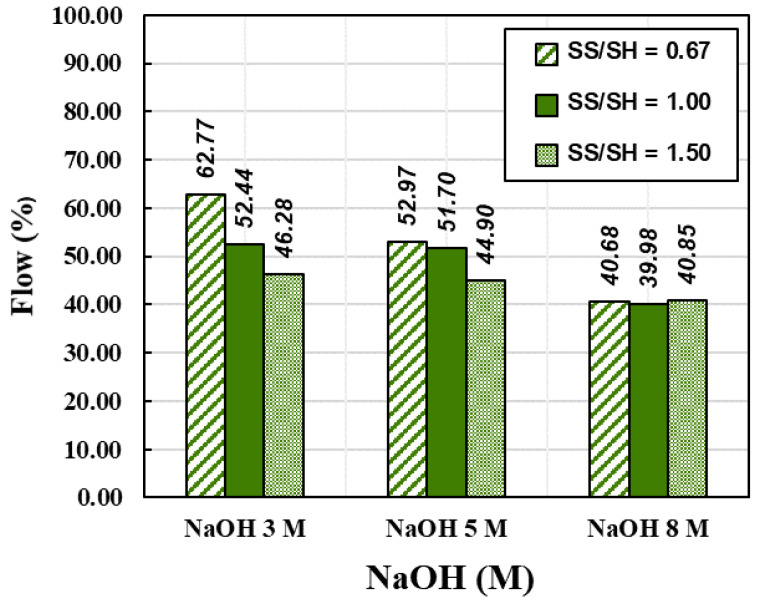
Flow values of various NaOH concentrations and SS/SH ratios.

**Figure 10 materials-15-03181-f010:**
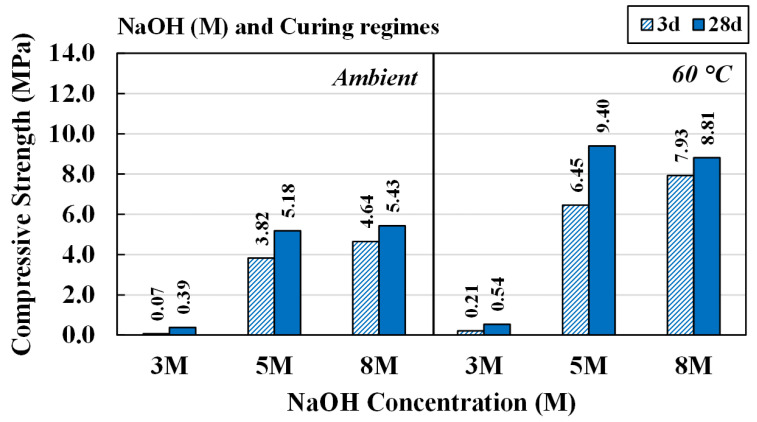
Compressive strength of AAM at various NaOH concentrations, cured at room temperature and at 60 °C.

**Figure 11 materials-15-03181-f011:**
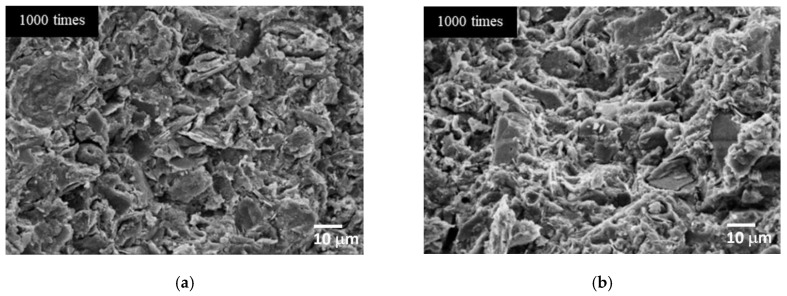
SEM images of the AAM preparing with 5 M NaOH at 28 days. (**a**) Cured at room temperature. (**b**) Cured in the oven at 60 °C.

**Figure 12 materials-15-03181-f012:**
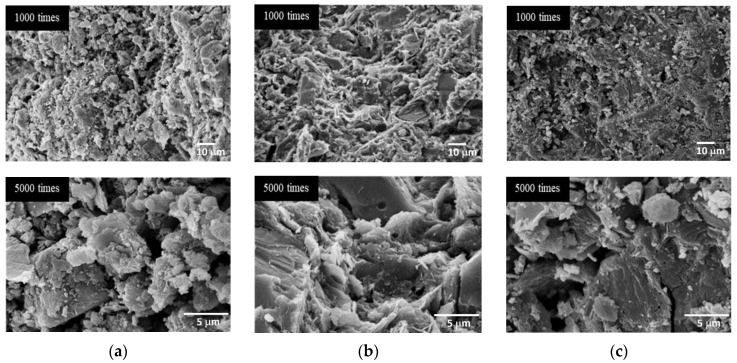
SEM images of the AAMs cured at 60 °C with various NaOH concentrations. (**a**) 3 M NaOH. (**b**) 5 M NaOH. (**c**) 8 M NaOH.

**Figure 13 materials-15-03181-f013:**
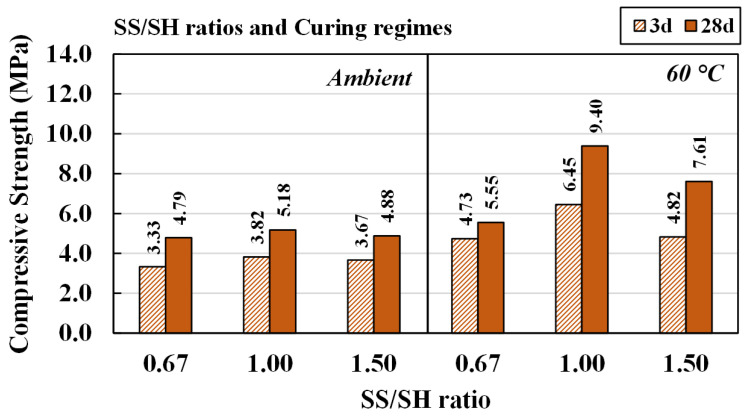
Compressive strength of AAM preparing with 5 M NaOH at various SS/SH ratios.

**Figure 14 materials-15-03181-f014:**
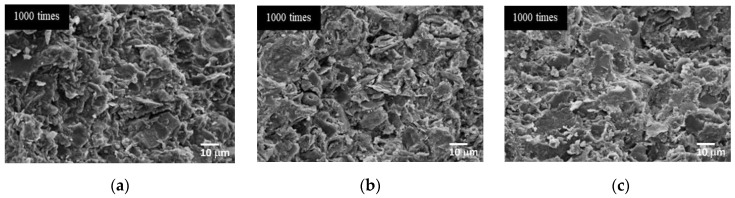
SEM images of the mixtures cured at room temperature with 5 M NaOH, an L/B ratio of 0.50, and various SS/SH ratios. (**a**) SS/SH = 0.67. (**b**) SS/SH = 1.00. (**c**) SS/SH = 1.50.

**Figure 15 materials-15-03181-f015:**
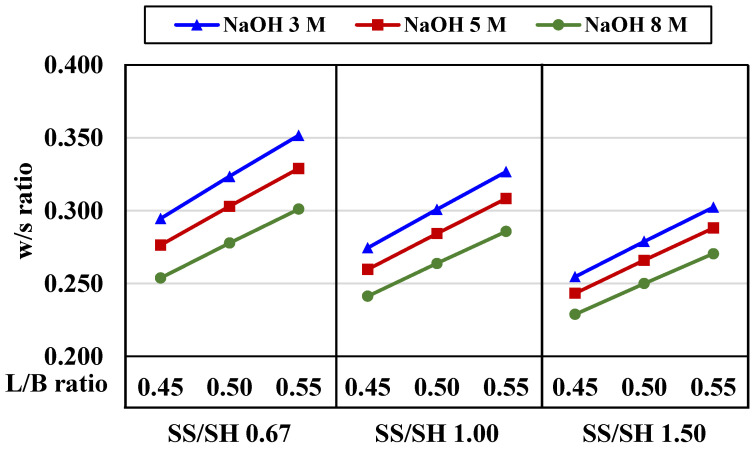
Water/solid (w/s) ratios compared to liquid alkaline/binder (L/B) ratios in CR-AAM systems.

**Figure 16 materials-15-03181-f016:**
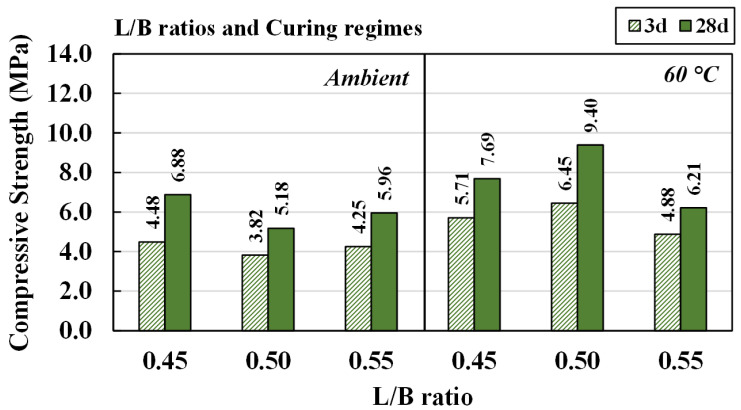
Compressive strength of mixtures at various L/B ratios, cured at room temperature and at 60 °C.

**Table 1 materials-15-03181-t001:** Chemical compositions of crushed rock.

Chemical Composition	Al_2_O_3_	SiO_2_	K_2_O	CaO	MnO	Fe_2_O_3_	TiO_2_
CR (%*w*/*w*)	8.06	21.99	4.39	59.53	0.28	5.74	0.57

**Table 2 materials-15-03181-t002:** Mixture designation of crushed rock-based AAM.

NaOH (Molar)	Crushed Rock (g)	SS/SH Ratio	L/B Ratio	NaOH Sol. (g)	Na_2_SiO_3_ Sol. (g)
3	1500	0.67	0.45	404.2	270.8
		0.67	0.50	449.1	300.9
		0.67	0.55	494.0	331.0
		1.00	0.45	337.5	337.5
		1.00	0.50	375.0	375.0
		1.00	0.55	412.5	412.5
		1.50	0.45	270.0	405.0
		1.50	0.50	300.0	450.0
		1.50	0.55	330.0	495.0
5	1500	0.67	0.45	404.2	270.8
		0.67	0.50	449.1	300.9
		0.67	0.55	494.0	331.0
		1.00	0.45	337.5	337.5
		1.00	0.50	375.0	375.0
		1.00	0.55	412.5	412.5
		1.50	0.45	270.0	405.0
		1.50	0.50	300.0	450.0
		1.50	0.55	330.0	495.0
8	1500	0.67	0.45	404.2	270.8
		0.67	0.50	449.1	300.9
		0.67	0.55	494.0	331.0
		1.00	0.45	337.5	337.5
		1.00	0.50	375.0	375.0
		1.00	0.55	412.5	412.5
		1.50	0.45	270.0	405.0
		1.50	0.50	300.0	450.0
		1.50	0.55	330.0	495.0

Remark: SS/SH = 0.55 for setting time and flow test and SS/SH = 0.50 for compressive strength test.

**Table 3 materials-15-03181-t003:** Peak area of XRD pattern of AAMs (25–35 two-theta degrees).

NaOH Concentration	RT Curing		Oven Curing	
	Peak Area (a.u.)	% Difference	Peak Area (a.u.)	% Difference
3 M	3816	0	3944	0
5 M	4205	10	4545	15
8 M	4445	16	4743	20

**Table 4 materials-15-03181-t004:** The w/s ratio of various alkaline dosages in CR-based AAM cement.

SS/SH Ratio	L/B Ratio	Water-to-Solid (w/s) Ratio		
		NaOH 3 M	NaOH 5 M	NaOH 8 M
0.67	0.45	0.295	0.276	0.254
	0.50	0.324	0.303	0.278
	0.55	0.352	0.329	0.301
1.00	0.45	0.275	0.260	0.241
	0.50	0.301	0.284	0.264
	0.55	0.327	0.308	0.286
1.50	0.45	0.255	0.243	0.229
	0.50	0.279	0.266	0.250
	0.55	0.302	0.288	0.271

## Data Availability

The data presented in this study are available on request from the corresponding author.

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
