# Peer review of "Properties and Microstructures of Crushed Rock Based-Alkaline Activated Material for Roadway Applications"

_materials, 2022, doi:10.3390/ma15093181_

Round 1

Reviewer 1 Report

The study presents results on using Ca-bearing crushed rock as a precursor in the alkali-activated geopolymerization process. There are a lot of relevant and useful results, English is fine and the text is clearly written. To introduce some improvements in the paper, remarks are the following:

- The title should state that Ca-bearing rock was used in the study.

- The Abstract section contains a high level of plagiarism, please rephrase.

- Section 2 should contain Materials and methods. All the methods employed, including methodology and the instrumentation used should be stated in this section.

- The XRD results of all the materials and products are supposed to be presented before any other result. Please, check the peaks in the corresponding XRD diagrams, and annotate all of them (especially in the lower and higher 2-theta regions). Besides, some of the peaks do not seem to be appropriately assigned. The disappearance of the main peak of quartz seems strange. Are you certain that aluminum is present in the oxide form, instead of maybe some mineral? In which form is the iron?

- The claim “It can be observed that a structural formation can be rapidly achieved at a very early age owing to the alkali reaction of high calcium content in CR.” should be more deeply discussed. Although high calcite content improves the geopolymerization process, the same is fastest in the beginning (in any material). Please, check more literature (https://doi.org/10.1016/j.clay.2022.106410). Also, in that (and other) papers it is already determined that the drying makes the geopolymerization process favorable, which should be mentioned in the text. Besides, the determined SS/SH ratio of 1 as optimal should be compared to the literature.

- It should be emphasized why the compressive strengths obtained are relatively low. Is it connected to the low portion of an amorphous phase in the precursor material? Is the formation of crystallinity and a small amorphous phase content a restriction for a greater compressive strength? You can also refer to the paper mentioned above.

Reviewer 2 Report

The manuscript entitled "Role of Alkaline Constituent on Properties and Microstructure of Crushed Rock Based Alkali-Activated Material for Low Strength Applications" presents an interesting experimental study conducted on the effect of NaOH molarity, SS to SH ratio and liquid to solid ratio on the compressive strength and microstructure of alkali activated materials. However, the paper has multiple issues that must be addressed. The paper needs minor revisions before it is processed further, some comments follow:

Title: The title is too long and unclear. Please consider replacing the title with a clear formula that reflects the content of the manuscript.

Introduction section

Figure 2- Please indicate the source.

Materials Section

The characterization of the raw materials should be exhaustive in order to provide all the relevant information that will assure the experiment's repeatability. Therefore, please specify the method and equipment used to evaluate the chemical composition. Also, please provide the particle size distribution and specific surface area of the raw material.

Table 1 - two types of iron oxides have been detected in this type of material, therefore, please replace Fe2O3 with FexOy or provide the scientific proof to support your results. Moreover, the XRD analysis from figure 12 doesn’t show the presence of phases with Fe content.

The Methods section is missing.

Each equipment and method have different principles and errors. Please briefly describe the parameters of the involved equipment and characterization techniques.

Experimental programs section

" left over night before demolding" – please provide the time.

Was the preparation method established in accordance with previous studies (cite those studies) or preliminary experiments (please introduce this statement)? How have been established the values for SS/SH, curing temperature, and NaOH molarity?

Results and discussions section

"SS/SH ratios were used at 0.67 and 1.50, while the NaOH concentrations were 3, 5, and 8 M." – why the authors didn’t evaluate the setting time and flow for the mixtures with SS/SH =1, is there any rationale in choosing only the lowest and the highest value?

Figure 5 - How many samples have been tested. Please provide the number of samples and the standard deviation values.

"An L/B ratio of 0.50 and an SS/SH ratio of 1.00 were set for curing the AAM at RT and 60 °C" – Why only this mixture wasn’t considered in 4.1. The information and data presented in 4.1 are not relevant for 4.2 and the experiments don’t provide the necessary information for the readers to understand the effect of the parameters on the properties of the obtained mixture.

Please introduce in 4.1. and 4.2. the results for all mixtures.

Figure 6, Figure 7  and Figure 9– Please introduce figure labels to highlight the areas of interest for the readers.

Figure 8 - How many samples have been tested. Please provide the number of samples and the standard deviation values.

XRD analysis – There are multiple peaks present in the XRD pattern which haven’t been considered. Why do the authors consider some peaks instead of others (there are some clear peaks around 6, 17, 24, 32,33,43, 46, 51 etc.)? Please improve the description of the XRD spectra. Also, the Fe containing phases are missing.

Reference section

The references are old. There are no references to studies published in 2022, and only 3 from 2021 and 3 from 2020. Please include more relevant and recent studies.

Round 2

Reviewer 2 Report

The authors considered most of my comments and the manuscript was improved accordingly. The paper can be published in the present form.